# Characterization of Atmospheric Fine Particles and Secondary Aerosol Estimated under the Different Photochemical Activities in Summertime Tianjin, China

**DOI:** 10.3390/ijerph19137956

**Published:** 2022-06-29

**Authors:** Jinxia Gu, Zexin Chen, Nan Zhang, Shitao Peng, Wenjing Cui, Guangyao Huo, Feng Chen

**Affiliations:** 1School of Science, Tianjin Chengjian Unversity, Tianjin 300384, China; chenzexin1105@163.com (Z.C.); zn45002124@163.com (N.Z.); cuiwenjing@tcu.edu.cn (W.C.); micheal@tcu.edu.cn (G.H.); 2Tianjin Research Institute for Water Transport Engineering, Tianjin 300456, China; 3School of Civil Engineering, North China Institute of Aerospace Engineering, Langfang 065000, China; chf_chendeng@nciae.edu.cn

**Keywords:** atmospheric fine particles, secondary aerosol, photochemical reaction

## Abstract

In order to evaluate the pollution characterization of PM_2.5_ (particles with aerodynamic diameters less than or equal to 2.5 μm) and secondary aerosol formation under the different photochemical activity levels, CO was used as a tracer for primary aerosol, and hourly maximum of O_3_ (O_3,max_) was used as an index for photochemical activity. Results showed that under the different photochemical activity levels of L, M, LH and H, the mass concentration of PM_2.5_ were 29.8 ± 17.4, 32.9 ± 20.4, 39.4 ± 19.1 and 42.2 ± 18.9 μg/m^3^, respectively. The diurnal patterns of PM_2.5_ were similar under the photochemical activity and they increased along with the strengthening of photochemical activity. Especially, the ratios of estimated secondary aerosol to the observed PM_2.5_ were more than 58.6% at any hour under the photochemical activity levels of LH and H. The measured chemical composition included water soluble inorganic ions, organic carbon (OC), and element carbon (EC), which accounted for 73.5 ± 14.9%, 70.3 ± 24.9%, 72.0 ± 21.9%, and 65.8 ± 21.2% in PM_2.5_ under the photochemical activities of L, M, LH, and H, respectively. Furthermore, the sulfate (SO_4_^2−^) and nitrate (NO_3_^−^) were nearly neutralized by ammonium (NH_4_^+^) with the regression slope of 0.71, 0.77, 0.77, and 0.75 between [NH_4_^+^] and 2[SO_4_^2−^] + [NO_3_^−^]. The chemical composition of PM_2.5_ was mainly composed of SO_4_^2−^, NO_3_^−^, NH_4_^+^ and secondary organic carbon (SOC), indicating that the formation of secondary aerosols significantly contributed to the increase in PM_2.5_. The formation mechanism of sulfate in PM_2.5_ was the gas-phase oxidation of SO_2_ to H_2_SO_4_. Photochemical production of nitric acid was intense during daytime, but particulate nitrate concentration was low in the afternoon due to high temperature.

## 1. Introduction

Since the serious haze incidents in 2013 in China, the regional atmospheric environment pollution, characterized by fine particulate matter (PM_2.5_; particles with aerodynamic diameters less than or equal to 2.5 micrometers (μm)), has attracted widespread attention, and serious haze pollution has endangered the public health [1,2,3,4]. In order to alleviate the severe situation of atmospheric environment pollution and effectively improve air quality, a series of control policies have been formulated, such as the Air Pollution Prevention and Control Action Plan in China [5]. Through the cooperative prevention and strict implementation of control policies in cities or regions, the mass concentration of PM_2.5_ has decreased on average in some representative regions, for example the annual mean mass concentration of PM_2.5_ is less than 35 μg/m^3^ in the Pearl River Delta (PRD) region, but the mass concentration of PM_2.5_ is still high in the Beijing–Tianjin–Hebei (BTH) and Yangtze River Delta (YRD) regions [5,6,7]. Furthermore, the mass concentration of ozone (O_3_) has greatly increased and the compound pollution caused by the interaction between O_3_ and PM_2.5_ has become the main problem of atmospheric pollution [8,9,10,11].

Many studies have been conducted on the spatio-temporal distribution characteristic of PM_2.5_ and O_3_ in China. In general, the atmospheric pollution of PM_2.5_ is heavier in autumn and winter [7,12,13], while the atmospheric pollution of O_3_ is more prominent in summer [14]. Although the serious pollution events owing to the PM_2.5_ or O_3_ usually occur in the different seasons, the mutual influence between the PM_2.5_ and O_3_ is always present, and is especially important in summer. As a photochemical oxidizer, the O_3_ oxidation is stronger with the higher mass concentration [15,16,17]. The secondary components of PM_2.5_ are formed by the oxidation of gasous precursors, which increase the oxidation character and promote the oxidation of the secondary components in atmospheric particulate matter [18,19]. Based on the analysis of the air quality observation data of Taipei, the estimation method of secondary aerosols in PM_10_ (the particle size less than or equal to 10 μm) is established under the different photochemical activities [20]. This method to estimate secondary aerosol has also been used in Pudong District of Shanghai in China, and shows that it is an effective method for estimating secondary aerosol using observation data of pollutants [19]. Due to the combined effect of emission source structure or intensity, geographical location and meteorological conditions, the complex relationship between PM_2.5_ and O_3_ is very different in different regions or cities. For example, the relationship between the meteorological conditions and PM_2.5_ and O_3_ concentrations has been studied in the BTH region [21]. The relationship between particulate matter and O_3_ is found in the North China Plain [9,11]. However, the information and relevant actions are still not sufficient to fully control the related pollution of O_3_ and PM_2.5_ because of the lack of the interaction analysis under different photochemical activities.

As the BTH region is one of the representative regions of economic development in China, its atmospheric quality has been receiving more and more attention. Tianjin is in the north-central part of the BTH region. As the BTH is the economic center of the Bohai region, its industry is developed. The main industries include petrochemical, electronics, machinery manufacturing and iron and steel metallurgy, among others. Tianjin is one of the cities with the most serious pollution of PM_2.5_. Air pollution prevention and control action plans have been implemented in Tianjin since 2013, and the PM_2.5_ mass concentrations decrease year by year [22]. However, PM_2.5_ pollution is still serious [23]. In this study, based on the continuous observation, the hourly maximum of O_3_ (O_3,max_) was used as an index to measure the levels of photochemical activity. CO was used as a tracer for primary emissions from natural origin sources such as sea salt and dust. Using the secondary aerosol estimation methods based on PM_2.5_/CO, the formation of secondary aerosols was estimated under different photochemical activities in summer 2020 in Tianjin, China. The main objective of the present work was to chemically characterize PM_2.5_ and to investigate the relationships between O_3_ and PM_2.5_ as well as their association with gaseous pollutants (NO_2_ and SO_2_) under the different photochemical activities.

## 2. Data and Methods

### 2.1. Sources of Monitoring Data

The air quality monitoring data are downloaded from the data center website of the China National Environmental Monitoring Center (http://www.cnemc.cn/sssj/) (accessed on 1 June 2021). The dataset contains the ambient mass concentrations of PM_2.5_, O_3_, NO_2_, CO and SO_2_, which are measured following the required equipment and method principles described in the technical specifications of the China National Environmental Monitoring Center (http://www.cnemc.cn/jcgf/) (accessed on 1 June 2021). PM_2.5_ is measured using a tapered element oscillating microbalance (TEOM) PM_2.5_ analyzer (Model RP1400a, Sunset Laboratory Inc., Tigard, OR, USA). NO_2_ is measured using a chemiluminescence NOx analyzer (Model 42C/I, Thermo-Fisher Scientific, Waltham, MA, USA), SO_2_ is measured using a pulsed fluorescence SO_2_ analyzer (Model 43C/I, Thermo-Fisher Scientific, Waltham, MA, USA), O_3_ is measured using a UV photometric O_3_ analyzer (Model 49C/I, Thermo-Fisher Scientific, Waltham, MA, USA), and CO is measured with a nondispersive infrared analyzer (Model 48i, TE). The data observed by the Tianjin monitoring site (39.08° N, 117.21° E) in the network is used from 1 June to 31 August 2020.

The continuous water-soluble inorganic ions and organic carbon (OC) and elemental carbon (EC) are measured on the top of the fourth floor (39°6′ N, 117°6′ E) in Tianjin Chengjian University, about 15 m from the ground. The main traffic line of Jinjing Road is the south and about 200 m away the sampling point, and the other surrounding areas are mainly residential communities. The water-soluble inorganic ions (i.e., SO_4_^2−^, NO_3_^−^, Cl^−^, NH_4_^+^, Na^+^, K^+^, Ca^2+^, and Mg^2+^) in PM_2.5_ are determined during the same observation period by the ambient ion monitor instrument (AIM, URG Corporation, Chapel Hill, NC, USA, CarURG9000B). OC and EC are measured Semi-Continuous OCEC instrument (Model 4, Sunset Laboratory Inc., Tigard, OR, USA). An automatic meteorological monitoring instrument (Milos520, Vaisala, Vantaa, Finland) is used to record the main meteorological parameters, including temperature, humidity, wind speed, and wind direction.

The PM_2.5_ data are initially available based on 5 min averages. The hourly means are calculated using a minimum of nine 5 min averages, and the daily means are calculated using a minimum of 18 1 h averages; otherwise, the hourly and daily value is considered missing. The other daily data (gaseous pollutants, water-soluble inorganic ions, OC, EC and meteorological parameters) are averaged over 24 h periods when at least 75% of hourly data are available for each day; the missing values are excluded from the analysis.

### 2.2. Classification of Photochemical Activity Levels

Because the formation of secondary aerosol is closely related to the photochemical activity, O_3_ is often used as an index of photochemical reactions [15,16,17,20,24]. To differentiate the effect of the different photochemical activity levels, in this study, photochemical activity is categorized into four groups (L, M, LH and H) based on hourly maximum of O_3_ mass concentration: L denotes O_3,max_ < 100 μg/m^3^ (8 h O_3_ standard in the National Ambient Air Quality Standard (NAAQS, GB3095-2012) of China; M denotes 100 μg/m^3^ ≤ O_3,max_ < 160 μg/m^3^ (8 h O_3_ standard in GB3095-2012); LH denotes 160 μg/m^3^ ≤ O_3,max_ < 200 μg/m^3^ (1 h O_3_ standard in GB3095-2012); H denotes O_3,max_ ≥ 200 μg/m^3^ (1 h O_3_ standard in GB3095-2012).

As the Kruskal–Wallis test does not assume normality in the data, it is much less sensitive to outliers than the one-way Analysis of Variance (ANOVA). The Kruskal–Wallis test is used to determine the statistically significant difference in PM_2.5_ and its chemical components under the different photochemical levels in this study.

### 2.3. Estimating the Photochemical Secondary Aerosols

In order to understand the formation of secondary aerosol in PM_2.5_, the secondary aerosol in PM_2.5_ is estimated occurring with the intense levels of photochemical activity. CO and O_3_ are used as indicating species for primary sources of motor vehicle emissions and secondary sources due to photochemical activity levels. When the level of photochemical activity is L, the observed PM_2.5_ mass concentration is mainly from primary source emissions (including sea salt). Using the hourly ratio of PM_2.5_/CO to represent the primary aerosol emission, the proportion of secondary components in PM_2.5_ is greater when the ratio of PM_2.5_/CO is greater and has been already clarified this hypothesis in [20,21]. The regression analysis between PM_2.5_ and CO is conducted and the correlation coefficients are 0.70, 0.55, 0.62, and 0.63 under the photochemical activity level of L, M, LH and H, respectively. This illustrates that photochemical activities are low when O_3,max_ is less than 100 μg/m^3^. The ratio of PM_2.5_/CO is a useful tracer for estimating primary aerosol when photochemical activity is high [20].

When the photochemical activity levels are M, LH, and H, the photochemical activity is relatively high, and the primary aerosol in PM_2.5_ is estimated by multiplying the ratio of PM_2.5_/CO by the CO. In this study, the primary aerosols are estimated using the hourly CO mass concentration under the different photochemical activity levels of M, LH and H. The equations used for calculation of the primary aerosol are as follows [16,25].
(1)$${{(\mathrm{PM}}_{2.5})}_{P,M,t}{=\mathrm{CO}}_{M,t}\times {{(\mathrm{PM}}_{2.5}/\mathrm{CO})}_{P,L}$$
(2)$${{(\mathrm{PM}}_{2.5})}_{P,\mathrm{LH},t}{=\mathrm{CO}}_{\mathrm{LH},t}\times {{(\mathrm{PM}}_{2.5}/\mathrm{CO})}_{P,L}$$
(3)$${{(\mathrm{PM}}_{2.5})}_{P,H,t}{=\mathrm{CO}}_{H,t}\times {{(\mathrm{PM}}_{2.5}/\mathrm{CO})}_{P,L}$$

In the aforementioned formula, P denotes primary pollutant; t denotes any hour of the day; L, M, LH, and H denots the different Photochemical activity levels; and (PM_2.5_/CO)_p,L_ is the 25th percentile (0.014) of the hourly ratio for PM_2.5_/CO at L photochemical activity.

The estimated secondary aerosol mass concentration is calculated by the observed PM_2.5_ mass concentration deducting the primary PM_2.5_ mass concentration, according to the following equations:(4)$${{(\mathrm{PM}}_{2.5})}_{\sec,M,t}={\left({\mathrm{PM}}_{2.5}\right)}_{\mathrm{obs,M,t}}-{{(\mathrm{PM}}_{2.5})}_{P,M,t}$$
(5)$${{(\mathrm{PM}}_{2.5})}_{\mathrm{sec,LH,t}}={\left({\mathrm{PM}}_{2.5}\right)}_{\mathrm{obs,LH,t}}-{{(\mathrm{PM}}_{2.5})}_{\mathrm{P,LH,t}}$$
(6)$${{(\mathrm{PM}}_{2.5})}_{\mathrm{sec,H,t}}={\left({\mathrm{PM}}_{2.5}\right)}_{\mathrm{obs,H,t}}-{{(\mathrm{PM}}_{2.5})}_{\mathrm{P,H,t}}$$

In the aforementioned formula, sec denotes secondary pollutant; obs denotes observed value; t denotes any hour of the day; and L, M, LH, and H denote the different photochemical activity levels.

The secondary organic aerosol (SOA) has been an important component of PM_2.5_ [26,27,28,29,30] and can be formed from the photochemical oxidation reactions of VOCs. In this study, the primary organic aerosol (POA) and secondary organic aerosol (SOA) are estimated by the previously reported ratios, POA/POC (=1.2 μg/μgC) and SOA/SOC (=2.2 μg/μgC) [31,32]. The mass concentrations of POC and SOC are estimated using the method of the minimal ratio of OC/EC [33]:(7)$${\left(\mathrm{POC}\right)}_{t}{=(\mathrm{EC})}_{t}\times {\left(\mathrm{OC}/\mathrm{EC}\right)}_{\mathrm{pri},\min}$$
(8)$${\left(\mathrm{SOC}\right)}_{t}{=(\mathrm{OC})}_{\mathrm{obs},t}-{\left(\mathrm{POC}\right)}_{t}$$

In the aforementioned formula, OC and EC are the measured hourly mass concentrations; t denotes any hour of the day; and (OC/EC)_pri,min_ is the minimum of OC/EC in primary emissions. At urban locations and in this study, the minimum OC/EC observed in ambient air is the absolute minimum of all the data in the time series and is used to represent (OC/EC)_pri_.

## 3. Results and Discussion

### 3.1. Mass Concentration of PM_2.5_ and O_3_

The mass concentrations of PM_2.5_ and O_3_ are shown in Figure 1 during the summertime 2020 in Tianjin, China. As shown in Figure 1a, the PM_2.5_ daily mean mass concentration was from 10.8 to 81.4 μg/m^3^, and the quarterly mean mass concentration 37.7 ± 19.8 μg/m^3^. Compared to the NAAQS of GB3095-2012, the PM_2.5_ daily mass concentration exceeded the class I (35 μg/m^3^) and class II (75 μg/m^3^) in 46 days and 1 day, respectively, accounting for 50% and 1% of the 92 days during the summertime 2020 in Tianjin, China. Compared to air quality guidelines (15 μg/m^3^) of the World Health Organization (WHO), during the observation period, the PM_2.5_ daily concentration exceeded 90 days and accounted for 97.8%. As shown in Figure 1b, the O_3_ seasonal mean concentration was 37.7 ± 19.8 μg/m^3^. The O_3_ daily 8 h average maximum (O_3,8h max_) mass concentration was between 58.6 and 274.3 μg/m^3^ during the summertime. According to the class I of O_3,8h max_ (100 μg/m^3^) and the class II of O_3,8h max_ (160 μg/m^3^), the O_3,8h max_ mass concentration, respectively, exceeded 46 and 34 days, accounting for 50% and 37% of the 92 days in the summertime 2020 in Tianjin, China. These indicated that the combined pollution of PM_2.5_ and O_3_ has been shown during the observation period in Tianjin, China.

The COVID-19 pandemic unexpectedly broke out at the end of 2019. Due to the highly contagious, widespread, and risky nature of this disease, pandemic prevention and control measures were adopted in China, such as the public staying at home and some industries shutting down, which had an impact on the structure of pollution sources and air quality. Therefore, it must be clarified that the results of this study were based on exceptional circumstances.

In order to access the significance differences in the PM_2.5_ under the different photochemical activity levels of L, M, LH and H. The Kruskal–Wallis test was performed and the *p* value was equal to 0.000 and was significantly smaller than 0.05, which proved the significant difference in the PM_2.5_ under the different photochemical activity levels of L, M, LH and H. In order to define the quantify of PM_2.5_ levels, under the different photochemical activities indexed by O_3_, the PM_2.5_ and O_3_ mass concentrations are shown in Figure 2. The PM_2.5_ mass concentration varied from 6 to 64 μg/m^3^, from 5 to 102 μg/m^3^, from 5 to 110 μg/m^3^, and from 10 to 130 μg/m^3^, and the mean values were 29.8 ± 17.4, 32.9 ± 20.4, 39.4 ± 19.1 and 42.2 ± 18.9 μg/m^3^ under the photochemical activity levels of L, M, LH and H, respectively, which indicated that the PM_2.5_ mass concentration increased with the strength of the photochemical activity as ascertained by O_3_ levels, and serious pollution of PM_2.5_ typically occurred in association with active photochemical processes or reactions.

### 3.2. Diurnal Variations in PM_2.5_

Based on the classification of photochemical activity, the diurnal profiles of hourly mean concentrations of PM_2.5_ and O_3_ are shown in Figure 3. The mass concentrations of PM_2.5_ and O_3_ presented a distinct diurnal pattern. The diurnal profile of PM_2.5_ showed a bimodal distribution. In contrast, the diurnal profile of O_3_ showed a unimodal distribution with peak value in the afternoon. When the photochemical activity level was L, both PM_2.5_ and O_3_ mass concentrations were relatively low and with small fluctuation amplitude, the maximum mass concentrations were 36.6 µg/m^3^ and 74.4 µg/m^3^ occurring at around 4:00 and 15:00, respectively. When the photochemical activity level was M, the mass concentrations of O_3_ were higher but the fluctuation amplitude was still low and the maximum value (121.5 µg/m^3^) was observed at 14:00. The fluctuation amplitude of PM_2.5_ was very significant. The maximum value (43.0 µg/m^3^) was observed at 7:00 and the minimum value (20.6 µg/m^3^) was observed at 18:00. When the photochemical activity level was LH, the mass concentrations of PM_2.5_ and O_3_ were obviously increasing during the daytime and the maximum values were 46.6 µg/m^3^ and 171.0 µg/m^3^ and observed at 7:00 and 16:00, respectively. When the photochemical activity level was H, the mass concentration of O_3_ further increased with a very significant fluctuation amplitude in the daytime and the maximum (211.0 µg/m^3^) was observed at 16:00. The PM_2.5_ mass concentrations also increased with small fluctuation amplitude in the daytime. The maximum mass concentration of PM_2.5_ increased to 49.3 µg/m^3^ and lagged until 9:00. In summary, under the different levels of photochemical activity, PM_2.5_ showed similar diurnal variation profile affected by the morning rush hours, the maximum appeared at around 5:00–9:00, after which the PM_2.5_ mass concentration gradually decreased and its minimum mass concentration occurred around 16:00–18:00 and then gradually increased affected by the night rush hours until 23:00. The mass concentration of O_3_ showed single peak pattern with lower mass concentration at night and rising rapidly around 7:00 accompanied by the enhancement of solar radiation, reaching the maximum values around 14:00–16:00 and then gradually decreasing until 23:00.

### 3.3. The Effect of Meteorological Parameters in PM_2.5_

Under the different photochemical activity levels, the bivariate polar plot (Figure 4) showed the PM_2.5_ as a function of wind speed and direction, which was used to identify sources responsible for the significant concentrations [32]. Under the L photochemical activity level, the mass concentrations of PM_2.5_ were high and associated with low wind speeds (<1.5 m/s) indicating the dominance of local pollutants. Under the photochemical activity levels of M, LH and H, the high mass concentrations of PM_2.5_ were associated with low wind speeds (<1.5 m/s) and high wind speeds (>1.5 m/s) attributed from locally produced and long-range-transported pollutants. The high concentrations of PM_2.5_ were observed in multiple directions, except for the north, when the wind speeds were low, indicating large heterogeneity in the emission sources. Additionally, the high mass concentrations of PM_2.5_ with high wind speeds over the photochemical activities of LH and H was associated with the breeze from the southwest and northeast directions.

The other meteorological parameters, including temperature and relative humidity (RH), were also shown in Figure 5. Under the different photochemical activity levels of L, M, LH and H, the diurnal patterns of ambient temperature were typical unimodal and ranged from 23.0 to 27.2 °C, 24.6 to 29.3 °C, 24.0 to 32.7 °C, and 24.7 to 35.3 °C, respectively. The diurnal patterns of RH anti-correlated with temperature. RH ranged from 60.7% to 77.0%, 51.7% to 68.2%, 39.4% to 70.0%, and 31.1% to 64.8%, respectively. This indicated that the photochemical activity level increased along with temperature and conversely decreased with the increase in RH.

### 3.4. Secondary Aerosols Estimation

Equations (1)–(3) were used to estimate the primary mass concentrations of PM_2.5_ for the three O_3,max_ intervals of M, LH, and H. As shown in Figure 6, at the different photochemical activity levels, the diurnal patterns of estimated primary PM_2.5_ mass concentration were similar with a small fluctuation amplitude. The maximum values appeared at about 8:00 corresponding to the morning rush hours and the minimum values appeared around 15:00–16:00 owing to the joint effect of increasing of the mixing layer height (MLH) and atmospheric temperature. The primary PM_2.5_ concentrations then increased after 18:00, corresponding to the night rush hours, and then they were relatively steady during the night. Under the photochemical activity level of M, the ratio of estimated primary concentration to the observed mass concentration of PM_2.5_ (PM_2_._5,pri_/PM_2_._5,obs_) was about 0.30 from 0:00 to 5:00, then it gradually increased and reached a secondary peak value (0.38) at 11:00, lagging the morning rush hours by 2–3 h, then after a slight decrease it rapidly increased and reached the primary peak value (0.49) corresponding to the night rush hours. This indicated that diurnal variation in the primary PM_2.5_ mass concentration was mainly affected by traffic emission sources under the photochemical activity level M. Under the photochemical activity levels of LH and H, the ratio of PM_2_._5,pri_/PM_2_._5,obs_ presented a small variation (between 0.29 and 0.33), with no peaks related to the morning or night rush hours. It could be seen that the proportion trend under the M photochemical level was significantly different from under the photochemical levels of LH and H. Analyzing the reasons for this, it was possible that the secondary aerosol concentrations were lower at the photochemical level of M than that at the photochemical levels of LH and H (see Figure 7). Furthermore, the primary aerosol concentrations also varied at the different photochemical levels. The joint effect led to the different proportion trend.

The secondary PM_2.5_ concentrations were estimated using Equations (4)–(6) by subtracting the estimated primary PM_2.5_ from the observed PM_2.5_, as shown in Figure 7. At the M, LH and H photochemical activity levels, the daily variation trend of the generated secondary aerosols was nearly similar and between 26.0 and 32.2 μg⸳m^−3^ from 0:00 to 5:00, after which the mass concentration of the secondary aerosol significantly decreased at the M photochemical activity and reached the lowest value (10.6 μg/m^3^) at 18:00, after which there was an increase. Under the LH photochemical activity, the secondary aerosol reached the first peak (33.4 μg/m^3^) at 7:00, then gradually decreased and reached the lowest value (23.2 μg/m^3^) at 17:00, and no significant variations were registered after that. Under the H photochemical activity level, the secondary aerosol reached the first peak (34.7 μg/m^3^) at 9:00, then gradually decreased and reached the lowest value (25.9 μg/m^3^) at 17:00, and after that then slightly increased and reached the second peak value (32.0 μg/m^3^). This showed that the mass concentration of secondary PM_2.5_ aerosol increased with the increase in photochemical activity. The contribution ratio of the secondary PM_2.5_ to the observed PM_2.5_ was more than 58.6% at any hour under the photochemical activities of LH and H.

Because the national standards used by different countries are different, there were certain differences in the specific values selected when defining the photochemical activity level of O_3_, so there will be differences in the estimated primary aerosol and secondary aerosol. Therefore, there are certain limitations when comparing and analyzing different literatures.

The diurnal patterns of primary organic aerosol (POA) and secondary organic aerosol (SOA) are shown in Figure 8. This indicates that the daily variation in the POA mass concentration was from 0.76 to 1.42 μg/m^3^ under different photochemical activity levels, with the maximum value occurring at around 7:00 according to the traffic morning rush hour, implying that large amounts of hydrocarbons were emitted from vehicles and were beneficial to the formation of POA. The first and second peaks of the SOA mass concentration appeared at about 11:00–13:00 and 19:00–21:00, lagging about 4 h after morning and night rush hours, respectively. Since the secondary organic aerosol was formed from gaseous organic precursors, SOA was thus an important source of PM_2.5_ in Tianjin during this measurement period. This phenomenon was observed in the late summer 2002 in Pittsburgh [29] and in Mexico City Metropolitan Area in 2006 [33].

### 3.5. PM_2.5_ Chemical Characterization

The main chemical components, including water-soluble inorganic ions, OC, and EC in PM_2.5_, are shown in Table 1 under the different photochemical activity levels. The total mass concentration of the detected 10 components were 21.4 ± 12.4, 22.5 ± 15.9, 27.6 ± 14.6 and 26.4 ± 13.2 μg/m^3^, respectively accounted for 73.5 ± 14.9%, 70.3 ± 24.9%, 72.0 ± 21.9%, and 65.8 ± 21.2% in PM_2.5_ under the photochemical activity levels of L, M, LH, and H. The missed mass could be mainly non-soluble crustal materials and metal species. The statistical analysis of main chemical components was performed by the Kruskal–Wallis test and the *p* values were smaller than 0.05, indicating the significant difference under the different photochemical activity levels.

The diurnal profiles of the measured PM_2.5_ and chemical components reconstructed PM_2.5_ mass concentrations correlated well in Figure 9a showed that the diurnal variations in chemical components were steady, except for slightly elevated OC in the morning which might be resulted from vehicle emissions according to the morning rush hour. The diurnal variation in chemical components in Figure 9b was similar to Figure 9a. However, the mass concentrations of components were slightly enhanced. Figure 9c shows that the mass concentrations of chemical components were even higher and peak value of OC was shifted to 11:00, which was closer to the occurrence time of SOC by higher O_3,max_ (see Figure 7). For the photochemical activity of H, Figure 9d showed that the peak value of the sum over aerosol chemical components occurred at 11:00; however, the peak value of OC and SO_4_^2−^ reaches the summit at 13:00. This was due to the decrease in NO_3_^−^ after 11:00 which might result from evaporation loss of NO_3_^−^ at high ambient temperature [34,35,36,37]. The sum of aerosol chemical components was apparently enriched under the photochemical activity level H as shown in Figure 9d. This provided the evidence of secondary aerosol formation under high photochemical activity in Tianjin. The peak values of OC, SO_4_^2−^, and NO_3_^−^ were 5.0, 11.0, and 7.3 μg/m^3^, respectively, under the photochemical activity level of H, which was higher than that of 3.1, 9.4, and 5.6 μg/m^3^, respectively, under the photochemical activity level L. In summary, under the different photochemical activities, the peak value of PM_2.5_ was accompanied with the increase in the sum of mass concentrations of SO_4_^2−^, NO_3_^−^ and NH_4_^+^ (SNA) and OC. The hourly mass concentration maximum value of SNA in PM_2.5_ were 21.7, 24.9, 26.0 and 24.0 μg/m^3^ under the photochemical activity levels of L, M, LH, and H, respectively. This indicated that the formation of secondary aerosols contributed significantly to the increase in PM_2.5_ mass. Furthermore, as shown in Figure 10, under the photochemical activities of L, M, LH, and H, SO_4_^2−^ and NO_3_^−^ were nearly neutralized by NH_4_^+^, with a regression slope of 0.71, 0.77, 0.77, and 0.75 between 2[SO_4_^2−^] + [NO_3_^−^] and [NH_4_^+^], respectively, where [NH_4_^+^], [SO_4_^2−^], [NO_3_^−^] refers to the molar concentration (μmol/m^3^), suggesting that ammonia was abundant and that the formation of particulate nitrate and sulfate was under an ammonia-rich condition in this measurement period. The result was observed at a rural site in eastern Yangtze River Delta of China [38].

### 3.6. Insights into the Formation of SO_4_^2−^ and NO_3_^−^

In order to investigate the formation mechanism of SO_4_^2−^, the diurnal profiles of SO_4_^2−^ and SO_2_ mass concentration are shown in Figure 11 under the photochemical activity levels M, LH and H. The mean mass concentration of SO_4_^2−^ showed a rapid increase from 5.2 μg/m^3^ to 6.3 μg/m^3^ between 6:00 and 11:00. SO_2_ increased sharply from 6:00 and peaked at 10:00. The SO_2_ peak was thus 1 h earlier than the SO_4_^2−^ peak, suggesting rapid formation of secondary sulfate, from the increased concentrations of SO_2_ probably due to traffic emissions in the morning rush hours, coupled with the propitious photochemical conditions. It suggested that active oxidation of SO_2_ in the gas phase was mainly responsible for the observed morning production of sulfate. The oxidation of SO_2_ to sulfuric acid (H_2_SO_4_) could occur both in the aqueous phase and in the gas phase [39,40,41]. Recent studies have shown that the gas-phase oxidation of SO_2_ to H_2_SO_4_ had become more and more important [42]. In this pathway, SO_2_ was first oxidized by OH to sulfur trioxide (SO_3_) and then to H_2_SO_4_ [42,43].

In order to investigate the different formation mechanisms of nitrate, the daytime was divided into four stages, as shown in Figure 12: Stage I, 0:00–5:00; Stage II, 5:00–9:00; Stage III, 9:00–18:00; Stage IV, 18:00–23:00. During Stage I, the mass concentration of NO_2_ kept on slowly increasing while the temperature was relatively low and RH was relatively high. Moreover, the temperature and RH stayed relatively constant during this stage, as could be seen in Figure 5, implying that the thermal equilibrium entered a steadier state. Some previous studies have shown that nitrate radical (NO_3_) and/or N_2_O_5_ were converted to nitric acid (HNO_3_) by hydrolysis under high RH [44,45]. During Stage II, the mass concentration of NO_2_ rapidly increased and peaked at about 7:00, due to the traffic emissions and relatively low mixed layer height (MLH). At the same time, the mass concentration of O_3_ began to rise due to increasing solar radiation and more active photochemical reactions [46,47]. These observations together indicated that elevated nitrate concentration was primarily due to gas-phase photochemical oxidation. The result was consistent with the observation in Hong Kong [48]. During Stage III, the mass concentrations of NO_3_^−^ sharply decreased, probably because the NO_3_ radical was easily destroyed by photolysis under the strong photochemical activity of this period in the day. Furthermore, the decrease in the nitrate concentration could also be attributed to the evaporation of ammonium nitrate due to the increasing temperature and the enhancement of the atmospheric diffusion capacity as the MLH ascends [49]. During Stage IV, the temperature decreased while RH gradually increased (Figure 5). The mass concentration of O_3_ gradually decreased (Figure 3) and remained relatively high during the period. The mass concentration of NO_2_ rose after sunset due to traffic emissions during the evening rush hours coupled with lower MLH. Due to the presence of abundant NO_2_ and O_3_, as well as the absence of sunlight, the conditions were created for the concentration of NO_3_ radical to increase gradually [50], leading to a steady increase in NO_3_^−^ concentration in this period. This observation was consistent with the measurement in August 2011 in the urban area of Shanghai [51].

## 4. Conclusions

Based on the atmospheric environment monitoring data, the study has performed the chemical characterization of PM_2.5_ and the estimation of the related secondary aerosol during the summertime 2020 in Tianjin, China. Under the photochemical activities of L, M, LH, and H, the PM_2.5_ mass concentration increased with the strength of the photochemical activity as ascertained by O_3_ levels and the diurnal patterns were similar to the maximum and minimum appeared at around 7:00–9:00 and 16:00, respectively. The diurnal patterns of estimated secondary aerosol increased with the strength increase in photochemical activity. In particular, the ratio of estimated secondary aerosol to the observed PM_2.5_ was more than 58.6% at any hour under the photochemical activity of LH and H, indicating secondary aerosols became the key issue of PM_2.5_ pollution in Tianjin. The chemical composition, including water-soluble ions, OC and EC, respectively accounted for 73.5 ± 14.9%, 70.3 ± 24.9%, 72.0 ± 21.9%, and 65.8 ± 21.2% of the observed PM_2.5_ mass. In particular, the SNA significantly contributed to the increase in PM_2.5_ under an ammonia-rich condition in this observation period. The photochemical oxidation of SO_2_ to H_2_SO_4_ was enhanced by the strong atmospheric photochemical reactions. Nitrate was mainly produced by the photochemical oxidation reactions in the daytime, but high temperature and low RH shifted the gas-to-particle partitioning of NH_4_NO_3_ to evaporation, thus led to an extremely low particulate nitrate concentration in the afternoon.

## Figures and Tables

**Figure 1 ijerph-19-07956-f001:**
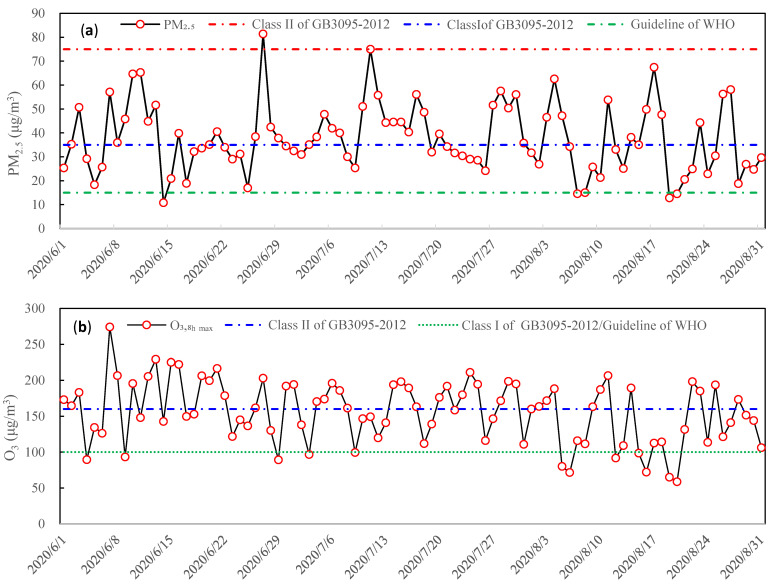
The mass concentrations of PM_2.5_ and O_3_ during summertime in Tianjin, China. (**a**) denotes PM_2.5_ mass concentration. (**b**) denotes O_3_ mass concentration.

**Figure 2 ijerph-19-07956-f002:**
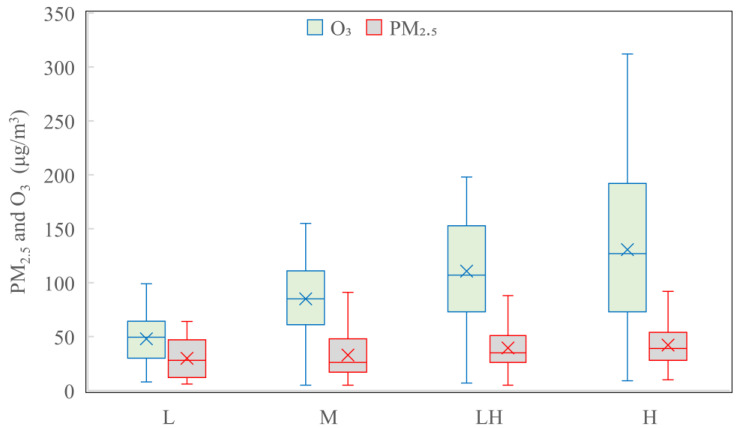
Box plots of PM_2.5_ and O_3_ mass concentrations under different photochemical activity levels during summertime 2020 in Tianjin, China. Box represents 25–75th percentiles, whiskers represent the minimum and maximum values except outliers, and horizontal lines in the middle of the boxes represent the median values.

**Figure 3 ijerph-19-07956-f003:**
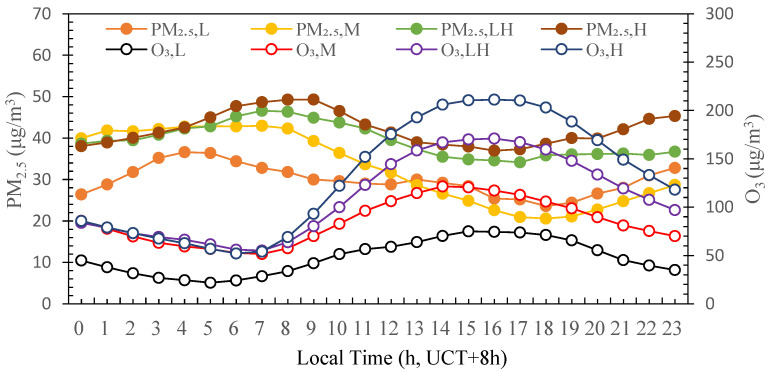
Diurnal variations in mean hourly concentrations of PM_2.5_ and O_3_ under different photochemical activities during the summertime 2020 in Tianjin, China.

**Figure 4 ijerph-19-07956-f004:**
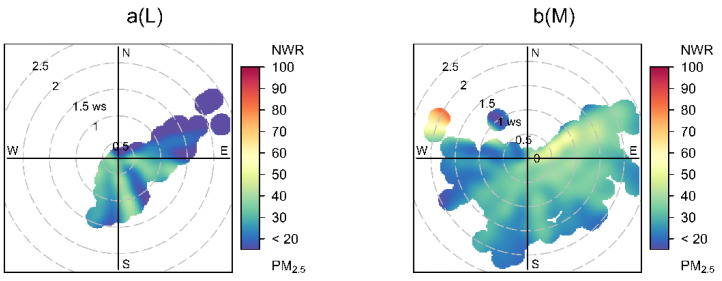
Bivariate polar plot of PM_2.5_ mass concentration (µg/m^3^) as a function of wind speed and direction under the different photochemical activity levels. The center of each plot represents a wind speed of zero, which increases radially outward. The concentration of PM_2.5_ is shown by the color scale. NWR represents non-parametric wind regression statistical approach. (**a**–**d**) denotes the different photochemical activity levels of L, M, LH and H.

**Figure 5 ijerph-19-07956-f005:**
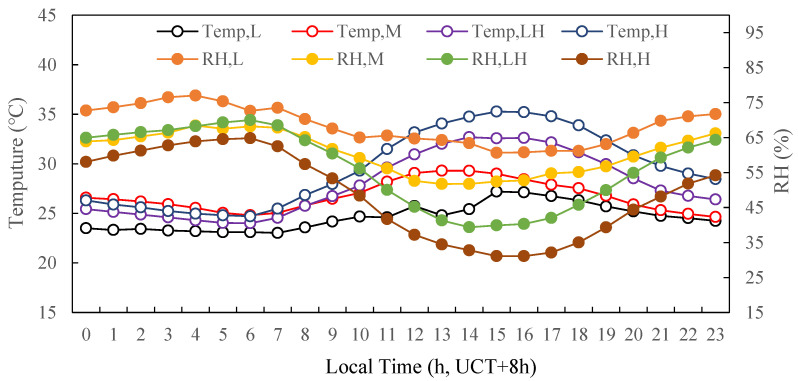
Diurnal variations in ambient temperature and relative humidity under the different photochemical activity levels.

**Figure 6 ijerph-19-07956-f006:**
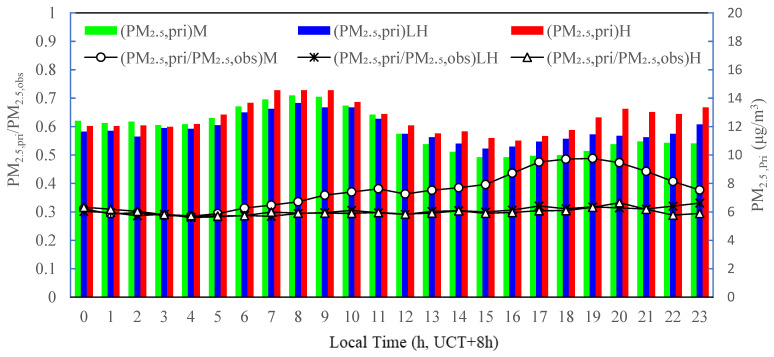
Diurnal variations in estimated primary PM_2.5_ mass concentrations and ratio of primary to observed PM_2.5_ mass concentrations at different photochemical activity levels in the summertime 2020 in Tianjin, China.

**Figure 7 ijerph-19-07956-f007:**
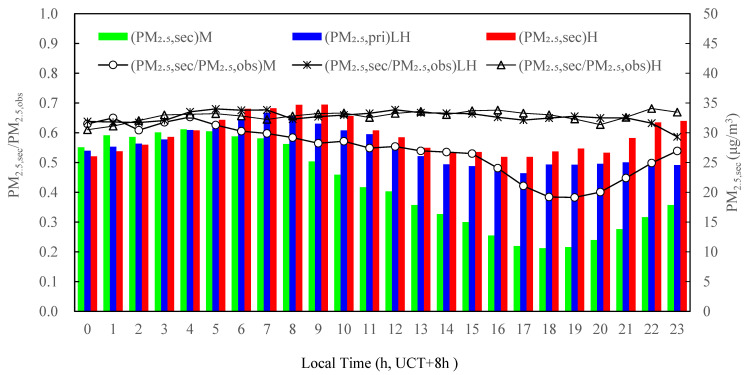
Diurnal variations in estimated secondary PM_2.5_ mass concentrations and ratio of secondary to observed PM_2.5_ mass concentrations at different photochemical activity levels in the summertime 2020 in Tianjin, China.

**Figure 8 ijerph-19-07956-f008:**
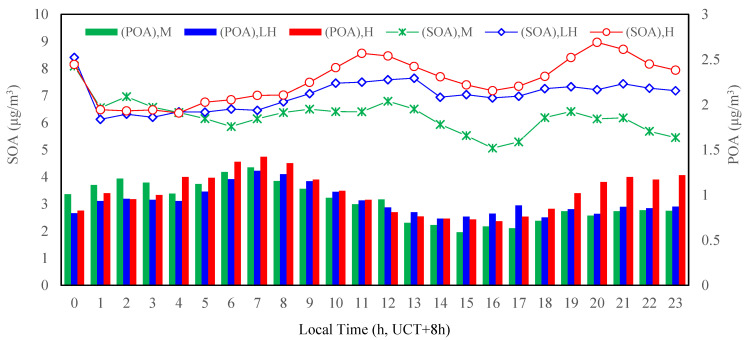
Diurnal variations in primary organic aerosol (POA) and secondary organic aerosol (SOA) at different photochemical activity levels in the summertime 2020 in Tianjin, China.

**Figure 9 ijerph-19-07956-f009:**
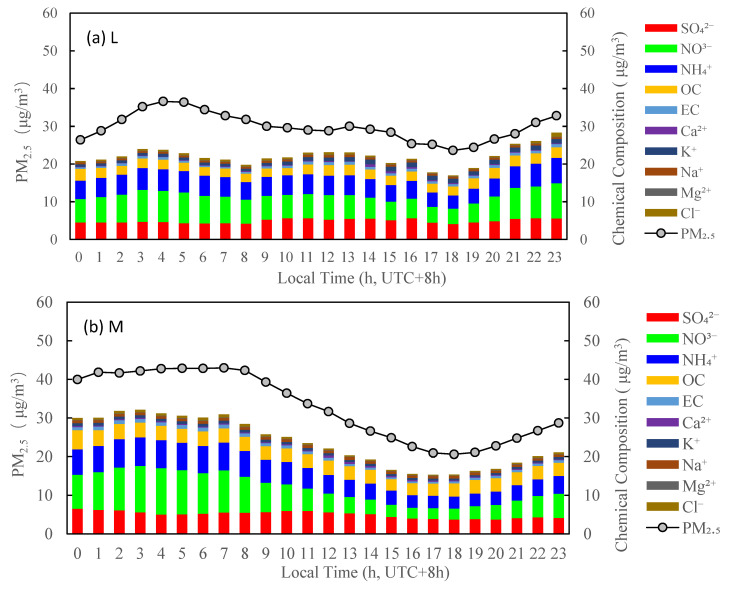
Diurnal patterns of PM_2.5_ and its chemical components concentrations under the different photochemical activity levels, in summertime 2020 in Tianjin, China. (**a**–**d**) denotes the different photochemical activity levels of L, M, LH and H.

**Figure 10 ijerph-19-07956-f010:**
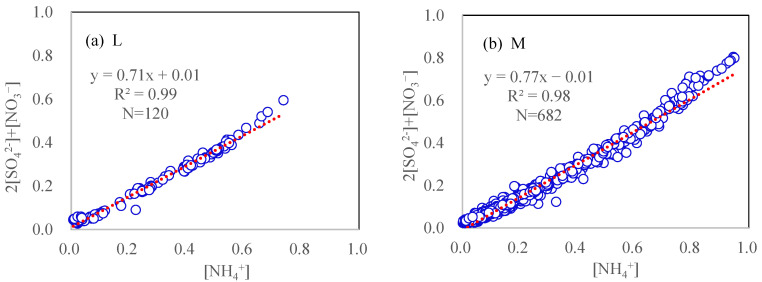
Scatter plots of 2[SO_4_^2−^] + [NO_3_^−^] vs. [NH_4_^+^] (μmol/m^3^) under the different photochemical activity levels in the summertime 2020 in Tianjin, China. (**a**–**d**) denotes the different photochemical activity levels of L, M, LH and H.

**Figure 11 ijerph-19-07956-f011:**
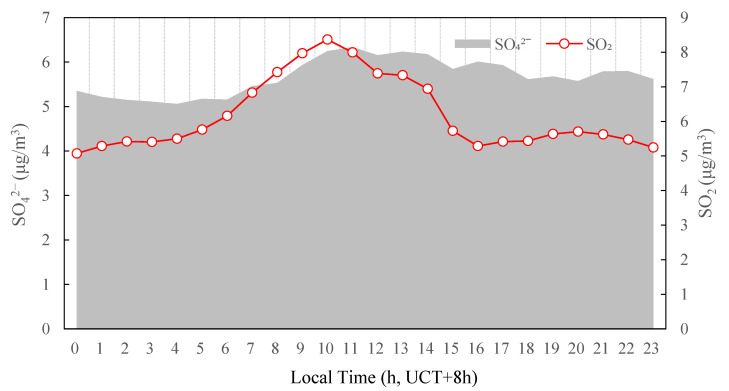
Diurnal variation in SO_4_^2−^ and SO_2_ concentrations averaged under the photochemical activity levels M, LH, and H in the summertime 2020 in Tianjin, China.

**Figure 12 ijerph-19-07956-f012:**
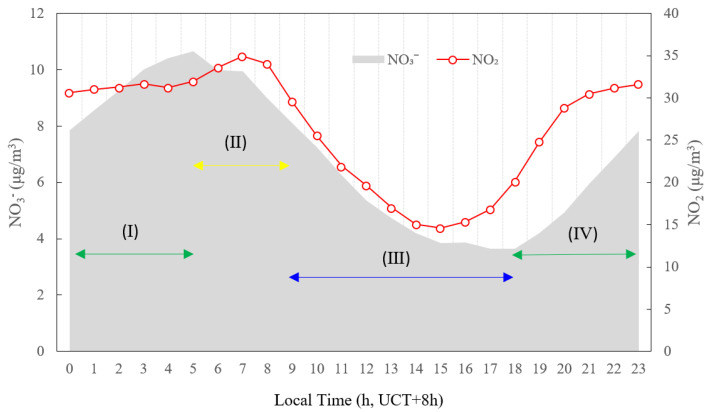
Diurnal variation in NO_3_^−^ and NO_2_ concentrations averaged under the photochemical activity levels M, LH and H, in summertime 2020 in Tianjin, China. Stage I, 0:00–5:00; Stage II, 5:00–9:00; Stage III, 9:00–18:00; Stage IV, 18:00–23:00.

**Table 1 ijerph-19-07956-t001:** The chemical component concentrations in PM_2.5_ under the different photochemical activity levels (mean ± SD, μg/m^3^).

Levels	OC	EC	Ca²⁺	K⁺	NH₄⁺	Na⁺	Mg²⁺	Cl^−^	NO₃^−^	SO₄²^−^	Sum
L	2.6 ± 0.8	0.7 ± 0.4	0.24 ± 0.34	0.49 ± 0.80	5.09 ± 3.65	0.51 ± 0.15	0.10 ± 0.07	0.44 ± 0.39	6.63 ± 5.26	4.93 ± 3.29	21.4 ± 12.4
M	3.7 ± 1.7	0.7 ± 0.5	0.32 ± 0.56	0.59 ± 0.48	5.26 ± 4.10	0.68 ± 0.39	0.13 ± 0.07	0.44 ± 0.47	6.81 ± 7.56	5.01 ± 3.35	22.5 ± 15.9
LH	4.0 ± 1.5	0.8 ± 0.5	0.47 ± 0.55	0.89 ± 0.53	6.42 ± 3.74	0.87 ± 0.39	0.18 ± 0.08	0.51 ± 0.49	7.68 ± 6.90	6.40 ± 3.54	27.6 ± 14.6
H	4.3 ± 1.6	0.9 ± 0.5	0.57 ± 0.57	0.77 ± 0.54	5.78 ± 3.61	0.76 ± 0.58	0.17 ± 0.08	0.55 ± 0.69	6.59 ± 6.29	6.36 ± 3.74	26.4 ± 13.2

## Data Availability

The data presented in this study are available on request from the corresponding author.

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
