# Peer review of "Characterization of Atmospheric Fine Particles and Secondary Aerosol Estimated under the Different Photochemical Activities in Summertime Tianjin, China"

_ijerph, 2022, doi:10.3390/ijerph19137956_

Round 1

Reviewer 1 Report

Minor revision

Author Response

Dear Reviewer:

Thanks for your attention and valuable advices. Based on your suggestion, we have revised our article. Please see the attachment for the specific modifications and answers.

Reviewer 2 Report

Dear Authors,

Please consider the following comments as constructive contributions.

This paper presents some methodological inconsistencies and weaknesses in data analysis, which prevent publication.

Furthermore, this paper needs extensive corrections of English language. Please consider the corrections identified in the attachment (it is not possible to reproduce these numerous language corrections needed here).

I-Methodological aspects to address:

1-This study is relative to June 1 to August 31 2020, but nothing is said about the pandemic. Was Tianjin in a lockdown? A contextualization is essential (and should be presented early in the paper).

How were the PM2.5 and O3 values affected by the pandemic mitigating measures? A comparison with previous 5-10 years would be very important, in order to ascertain in what extent the results/conclusions of this study are representative.

2-This study should be performed not only in three months (furthermore of the same season), but along the year (in the several seasons). That would allow to identify more significant differences in photochemical activity and other meteorological factors (e.g. between winter and summer) and thus to be able to draw more robust conclusions on the influence of the photochemical and meteorological conditions on the formation of secondary PM2.5. Mean monthly levels of PM2.5 and O3 should have been presented.

3-The O3 concentrations depend not only on solar radiation, but also on other meteorological factors (such as temperature, relative humidity and precipitation quantity), as well as to the intercontinental transport of O3 and its precursors (NOx and NMVOCs). For this reason, although it is valid to consider O3 concentration as a proxy to photochemical activity, it should also be referred in the Discussion that other factors, such as the variations in precursors emissions and long range transport of O3, also influence O3 concentrations.

4-For the definition of photochemical activity levels, the Authors have adopted ranges of O3 concentrations that are different from the ranges adopted in ref. 20; for example, in ref. 20 the L level corresponded to O3, max<60 ug.m-3 (and not to O3,max<100 ug.m-3).

These tier values are not absolute (they were adopted from legislation), but in order to somehow validate them, the Authors should perform a regression analysis between PM2.5 and CO, for each of the adopted photochemical activity levels (as in the Fig. 2 of ref. 20). For the L tier value to correctly represent an absent or very low photochemical activity, the correlation coefficient between PM2.5 and CO (assumed as a primary pollutant) should be high. For the other levels (M, LH, H), the correlation should become poorer, since secondary source becomes more representative.

5-According to ref 20, at the L level defined as O3, max<60 ug.m-3, there may be already some photochemical activity and formation of secondary aerosol. Using a L tier defined as O3, max<100 ug.m-3, the possibility of formation of secondary aerosol is higher, and thus this introduces an error in the estimation of primary aerosol under M, LH, and H photochemical conditions on the basis of (PM2.5/CO)P,L determined in that way. Consequently, the estimated secondary aerosol values are also affected by this error. This aspect must be addressed in the Discussion.

6-How did the Authors determine the (PM2.5/CO)P,L? It should be described and the value should be presented. Graphs of the diurnal profile of the ratio PM2.5/CO should be presented (at least in the Supplementary Information), for each of the photochemical activity levels (as in Fig. 5 of ref. 20), since these allow for the determination of (PM2.5/CO)P,L.

7-The (OC/EC)prim,min value should be presented, since it is crucial for the determination of POC and SOC (and POA and SOA), eq. 7 and 8.

How was this (OC/EC)prim,min ratio determined? In line 166-168, Authors state that (OC/EC)prim,min was the minimum of OC/EC observed.  

For the EC tracer method to be correctly applied, the data in the time series with the lowest 10% OC/EC ratio (assumed to be dominated by primary emissions) should be analyzed with linear regression (OC vs EC), to derive the site-specific primary OC/EC ratio (see ref. 31). This methodology should be applied.

8-The Authors should perform a statistical analysis of the significance of the differences between the values of PM2.5 (and O3) in the different classes (L, M, LH and H), in Figure 2. The non-parametric Mann–Whitney U test must be employed to assess significant differences between two independent groups and the Kruskal-Wallis test to compare two or more independent samples.

This analysis should also be performed for PM2.5secondary.

The Spearman correlation analysis between PM2.5, PM2.5secondary and O3 concentrations could also be performed.

9-The photochemical activity levels were defined on the basis of the O3 concentrations, as a proxy, so the factors that influence O3 concentrations should be studied. Spearman correlation analysis should be performed between O3 concentrations and: Temperature; RH; Wind speed; Precipitation quantity; and solar radiation (if available), since according to literature, these are factors that influence O3 concentrations, and consequently  the "photochemical activity levels" as defined in this work. 

10-Figure 7.: Would it be expected that the peak of PM2.5, secondary be at around 9:00? In ref. 20 and other works, the maximum was at 12:00-15:00, when solar radiation is more intense.

Furthermore, in the present study the maximum of O3 concentration occurred at around 16:00 (Fig. 3), but at this hour PM2.5, secondary is at a minimum.

Thus, in the present study, the maximum of PM2.5,secondary occurred before the maximum of O3 concentration. This result would not be expected. For example, in ref. 20, the maximum of PM2.5,secondary occurred 2–3 h latter than the maximum of ozone concentration (and the authors conjectured that the delayed time might correspond to the additional oxidation time needed to form condensable aerosol). These discrepancies from literature must be commented.

Please comment also the fact that the ratio PM2.5, secondary/PM2.5 presents almost no variation during the day, for the LH and H photochemical activity levels. (More insolated hours would be expected to present higher ratio values).

This profile is quite different from the profile presented in ref. 20 (Fig. 7d), where there are considerable variations of the ratio, from 0 to 30%, along the day. These different "behaviors" should be commented.

Finally, how can we explain that the PM2.5, secondary/PM2.5, observed ratio increases after 19:00, when there is no solar radiation, at the photochemical activity level M?

11-The average composition of PM2.5 should be presented, in terms of mean concentrations (+-s.d.) of each of the components, averaged for all the period (and all the photochemical activity levels). The percentage of SO42-, NO3-, NH4+, OC, EC (relative to PM2.5 mass) should be presented.

12-In section 3.5., boxplots such as in Figure 2, should be presented for NO3-, SO42-, NH4+, OC, EC, for the different photochemical activity levels.

Since SO42-, NO3- and NH4+ are secondary components in origin, a significant variation is to be expected with the photochemical activity levels. A statistical analysis should be performed (by Mann-Whitney or KW tests) in order to ascertain if the differences are significant.

This will be crucial to support the statements in lines 331-334, and in general to objectively compare the aerosol chemical composition under the different photochemical activity levels (avoiding vague statements as in line 332, "...the concentrations of aerosol components were slightly enhanced..." or in lines 340-341, ..."The sum of aerosol chemical components was apparently enriched when O3, max>200 ug/m3, as shown in Fig. 9(d)"...). This will allow to achieve the aim of the paper expressed in the title: to characterize the fine particles under different photochemical activities.

13-A correlation analysis between NO3-, SO42- and NH4+ versus O3 could also be performed .

II-More specific aspects to address:

Note: English language aspects are not included here (please consult the attachment file)

In all the paper: μg/m3 or μg.m-3 can either be adopted, but must be standardized throughout the document.

Lines 2-3: The title should be "Characterization of atmospheric fine particles and estimation of secondary aerosol under different photochemical activities in summertime in Tianjin, China"

Since only summertime is studied, this must be referred in the title so it reflects the real scope of the paper.

Line 11 and 37: attention to the definition of PM2.5 and to its units

Line 13: substitute "average" by "mean" (in all the paper)

Lines 31-31: There is no evidence of the mentioned specific reaction in the paper and thus the sentence should be removed.

Line 41: A reference should be inserted

Lines 44: The PM2.5 value refers to which year? 

Line 46-47: Where you say ..."the mass concentration of ozone (O3) has been greatly improved"..., you certainly mean "increased" (not improved), which has opposite meaning.

Line 57: References 15-17 are not related to China and the paragraph begining in line 50 is related to PM2.5 and O3 in China. Please choose more adequate references.

Lines 57-58: The sentence "The secondary components in PM2.5 were usually oxidized by precursors"... is not correct. It should be: "The secondary components of PM2.5 are formed by the oxidation of gaseous pollutant precursors."

Line 62: attention to the definition of PM10

Line : The sentence "CO was used as a tracer for primary emissions from natural origin sources such as motor vehicles,"... is obviously incorrect. 

It can be said: "CO can be assumed to be a nonreactive tracer for traffic, by neglecting its photochemical formation during air mass transport (ref. 20 and ref. therein).

Line 90: A subsection describing the Study site must be included.

Line 122: Standardize "photochemical activities" by "photochemical activity levels", in all the document.

Lines 127-130: The GB3095-2012 standard must be a bibliographic reference.

Line 134-135: A reference is mandatory here, to support the use of this methodology.

Line 137-138: It is not appropriate to consider the "PM2.5/CO ratio as a representation of the primary aerosol emission", since a higher ratio value does not correspond to higher primary aerosol emission (as recognized in the second part of the sentence). Please rephrase.

Line 141-142: In the sentence ...", the primary aerosol in PM2.5 was estimated by multiplying the ratio of PM2.5/CO by the CO mass concentration.", it must be specified that the referred PM2.5/CO ratio is indeed "(PM2.5/CO)P,L" and is determined under the lower photochemical activity conditions.

Line 145: A reference for these equations should be inserted here. 

Lines 159-161: It is stated that in this study, the primary organic carbon (POC) and secondary organic carbon (SOC) were estimated using the previously reported ratios POA/POC (1.2 ug/ugC) and SOA/SOC (2.2 ug/ugC), but the Authors are certainly refering to the calculation of POA and SOA.

Line 162: This method for the estimation of POC and SOC is known as the EC tracer method (ref. 31 and references therein).

Line 174-182: The PM2.5 daily mean concentrations should be compared not only to the GB3095-2012 standard, but also with the daily limit values of other international standards (e.g. Directive 2008/50/EC) and WHO daily guideline values. This contextualization is essential in terms of air quality and health risk.

Line 175: The meaning of the Classes (I and II) must be explicited. 

Line 181: order must be switched

Line 188: The meaning of the horizontal lines (class I and II thresholds) should be specified in the caption.

Lines 191-193: The information in lines 191-193, about the composition of the boxplot, should be presented in the Figure 2. caption. In the boxplot, mean values are represented by crosses and not squares (as stated).

Lines 194-196: There seems to be some data inconsistency, since these (maximum) values that are presented in the text do not seem to correspond to the values presented in Fig. 2 (please verify).

Figure 2.: The mean points in the graph should not be joined. 

Line 205: meteorological parameters are not shown in Fig. 3

Line 227: It must be said that this increase in PM2.5 concentrations is also the consequence of the rush hour in the end of the day.

Line 237: Ref. 32 must be wrong in this context. The reference to R package must be done in the Methodology section.

Figure 4.: Units are missing, both in PM2.5 concentration and wind speed.

The meaning of NWR must be specified in the caption.

Line 258-260: Rephrase (since it is not the photochemical activity level that makes the temperature  increase...): "This indicated that the photochemical activity level increased along with temperature"... "and conversely, higher photochemical activity levels were associated to lower relative humidities."

Line 261: This subtitle is not in the correct place

Line 264: Eqs. 1-3

Lines 266-267: The sentence in line 266-267 is true only for LH and H photochemical activity levels. 

Figure 6: Why is the daily profile different for M versus LH and H photochemical activity levels?

Line 287: Eqs. 4-6

Line 292: 24.7 ug.m-3 is not the minimum value at M photochemical activity level

Figure 7.: Fig.7 and Fig. 8 graphs have guidelines, but the other graphs do not have. The formating and graphical aspect of the graphs in the several Figures should be standardized.

Figure 7.: Would it be expected that the peak of PM2.5, secondary be at around 9:00? In ref. 20 and other works, the maximum was at 12:00-15:00, when solar radiation is more intense.

Furthermore, in the present study the maximum of O3 concentration occurred at around 16:00 (Fig. 3), but at this hour PM2.5, secondary is at a minimum.

Thus, in the present study, the maximum of PM2.5,secondary occurred before the maximum of O3 concentration. This result would not be expected. For example, in ref. 20, the maximum of PM2.5,secondary occurred 2–3 h latter than the maximum of ozone concentration (and the authors conjectured that the delayed time might correspond to the additional oxidation time needed to form condensable aerosol).

These discrepancies from literature must be commented.

Please comment also the fact that the ratio PM2.5, secondary/PM2.5 presents almost no variation during the day, for the LH and H photochemical activity levels. (More insolated hours would be expected to present higher ratio values).

Finally, how can we explain that the PM2.5, secondary/PM2.5, observed ratio increases after 19:00, when there is no solar radiation, at the photochemical activity level M?

Line 319: 3.5. PM2.5 chemical characterization

Line 320: …”measured by TEOM”… Methodological details should be presented in the Methodology section.

Line 329: I can not identify this trend of slightly elevated OC in the morning in Figure 9 (a). Instead, NO3- and EC concentrations seem to be higher in the morning and end of the day rush hours.

Lines 330-331: The profiles in Figure 9(a) and Figure 9(b) are not very similar...

Lines 333-334, 336 (and others): In all the paper the photochemical activity levels should always be referred to in the same way (L, M, LH and H, and not by the O3 concentration tiers).

Line 334-335: In Figure 9(c), the peak of OC at 11:00 is not apparent.

Line 336: This Fig. number is not correct

Line 337-338: By inspection of Figure 9(d), it is not true that the peak value of the reconstructed PM2.5 concentration occurred at 11:00.

Line 338: By inspection of Figure 9(d), it seems not to be true that the peak value of SO42- occurred at 13:00.

Lines 338-340: The statement in lines 338-340 is correct, but could be clarified.

I suggest: "At lower temperatures and higher relative humidity, NH4+ associates with NO3- due to the shifting of the NH4NO3(s) > HNO3(g) + NH3(g) equilibrium to the aerosol phase, and this contributes to the higher concentrations of NO3- in PM2.5 observed in the early morning. SO42- then tends to increase with enhanced solar radiation and photochemical activity, that promotes the formation of secondary sulphates (NH4)2SO4 (refs)."

Lines 340-342: The claimed higher level of reconstructed PM2.5 is not apparent from Figure 9(d) (at least when comparing with Figure 9(c)).

As previously mentioned, a statistical analysis (by the Mann-Whitney or KW tests), in order to ascertain if the differences of chemical component concentrations between the different photochemical activity levels are significant will be crucial, to make an objective comparison, avoiding these vague statements (..."the sum of aerosol chemical components is apparently enriched...").

Line 343: There seems to be some data inconsistency: from Figure 9(d), the peak concentration of NO3- seems higher than SO42-...

Line 348: It is interesting to note that under the photochemical activity level H, the SNA maximum value (24.0) was lower than under LH and M photochemical activities. This aspect must be addressed. Of course, more important than the maximum values, the mean (or median) values should be analyzed instead. But this result in fact demonstrates the need for a statistical test to determine if the differences are significant.

Line 349: Remove the word “fast” (the kinetic aspects of the secondary aerosol formation is not studied in this work)

Figure 9.: y axis label: "PM2.5 and components concentration (ug.m-3)" (in all graphs)

Figure 9(d): The EC color is different from the above graphs (standardize colors for each species in the several graphs)

Figure 10 (axis): units are missing.

The meaning of N should be indicated in the Figure caption.

This indication of N values (the number of data points considered in each photochemical activity level) is very important and should appear much sooner in the paper, in section 2.2.Classification of photochemical activity levels. Furthermore, N values should also appear in the first time averages of concentrations in each of the photochemical activity levels are calculated (in Figure 2).

Line 366: This claim of investigating the formation mechanisms of SO42-is an overstatement (the paper does not investigate mechanistic details).

Lines 388-390: Figure 12 does not support this statement, since the NO3- peak precedes the NO2 peak…

Furthermore, it makes no sense to refer the radical NO3, since it is a species that has not been measured or observed in this study.

Line 393: These references (46, 47) do not make sense here, but much earlier in the paper, in the Introduction, when the formation mechanism of O3 is first addressed.

Here, one should refer to the Figure 3, which shows the diurnal profile of O3.

Nevertheless, according to Figure 3, it is not true that the concentrations of O3 "rose quickly" during stage II (5:00-9:00).

Lines 393-395: Figure 12 does not support this statement of lines 393-395, since the NO3- peak precedes the NO2 peak!

Figure 12: The numbering of the stages shown in Figure 12 is not correct (it does not correspond to the numbering presented in lines 382-383, which makes more sense).

Conclusions: The Conclusions section should be restructured taking into account the comments and suggested amendments.

References: Please inser DOI in all references.

Author Response

Dear Reviewer:

Thanks for your attention and valuable comments. Based on your suggestion, we have revised our article. Please see the attachment for the specific modifications and answers.

Reviewer 3 Report

Manuscript ID: ijerph-1763044

Title; Characteristic of atmospheric fine particles and secondary aerosol estimated under the different photochemical activities in Tianjin, China

Although the topic is of interest to the Scientific community, before considering it for publication, this paper should be improved. Authors should reconsider the main objective of the paper according to the content. They should try to synthesize and emphasize the main findings of the study and avoid long sentences. Furthermore, authors should avoid drawing risky conclusions.

Evaluation; Major Revision.

1Abstract; Please revised the sentence “In order to evaluate the pollution characteristic of PM2.5 (particles with aerodynamic diameters less than 2.5 micrograms [μm])” either microgram or μm, only one for use all of the main text.

2Introduction; Line 63-66, Please provide the reference and evidence to support it. “However, in the actual atmosphere, coarse particles with a particle size greater than 2.5 μm were mainly contributed by primary emissions such as dust, which was not strongly related to photochemical activity, and the secondary aerosols generated by chemical reactions mainly appeared in PM2.5 (Ref.)?

3Sources of Monitoring Data

Why in this study use the OCEC analyses by NIOSH Method 5040? Actually, IMPROVE protocol is a widely use protocol for studying carbonaceous particles in Asia

4.Results and Discussion

The formation of secondary aerosols was estimated under different photochemical activities in the summer of 2020 in Tianjin, China. How about to concern about effects of the COVID-19 situation?

5.    Conclusion

The conclusions could be further developed, there is a lot of interesting data in the article.

Author Response

(The authors gave the same response as above.)

Round 2

Reviewer 2 Report

Dear Authors and Editor,

The revised version of the paper reveals improvement in the overall quality, but some aspects still need careful attention before publishing.

-The paper requires a thorough revision of English language before publishing, since many language incorrections have been introduced in this second version.

Other aspects also still need attention, e.g.:

-Figure 1: In the legend, "Class I" and "Class II" should appear instead of "75" and "35", and "WHO" instead of "15". The same correction is needed in Figure 1.(b), concerning O3 thresholds.

-Figure 2 (caption): Rephrase: "Box plots of PM2.5 and O3 mass concentrations under different photochemical activity levels during summertime 2020 in Tianjin, China. Box represents 25-75th percentiles, wiskers represent the minimum and maximum values except outliers and horizontal lines in the middle of the boxes represent the median values."

-Figure 6 (axis): "PM2.5pri/PM2.5obs", instead of "Ratio".

-Figure 7 (caption): "PM2.5sec/PM2.5obs", instead of "Ratio".

-Lines 332-333: Considering the values in Table 1, I can not reproduce the values of sum of the 10 components presented in lines 332-333 and respective percentages. Please verify.

-Line 335: "Kruskal-Wallis", instead of "KW". 

Furthermore, the Kruskal-Wallis test shoud have been mentioned in the "2. Data and Methods" section.

-The DOI of each reference should be inserted (it is a major added value for the reader).

Please see also the attachment "Coverletter with Reviewer comments":

-Reviewer comment 4: The regression analysis between PM2.5 and CO, in each photochemical activity level, could be presented in the Supplementary Information, and would greatly improve the paper.

-Reviewer comment 9: The Authors did not address this issue. It is essential to specify if Tianjin was in a lockdown during the sampling period and even if not, a reference to COVID-19 pandemic is essential, since it affected primary emissions and also had surprinsing effects in secondary pollutants, and thus, for future reference, the sampling period must contextualized. It must also be said in the Discussion that the results may be not representative of a "normal"/non-pandemic situation. 

-Reviewer comment 15: In line 185-186 it says: "The non-parametric Mann–Whitney U test between the PM2.5 and O3 was performed."

This must be rephrased, since the pertinent test was not between PM2.5 and O3 concentrations (of course), but between PM2.5 concentrations at different photochemical activity levels (and the same for O3), and the pertinent test in this situation is Kruskal-Wallis.

-Reviewer comment 21: A regression analysis between NO3-, SO42- and NH4+ versus O3 has not been performed and it would value the article.

Best regards

Reviewer 3 Report

This revised manuscript is suitable for publication.

Author Response

Thank you very much for your kind comments